# Identification of the Microbiome Associated with Prognosis in Patients with Chronic Liver Disease

**DOI:** 10.3390/microorganisms12030610

**Published:** 2024-03-19

**Authors:** Kenta Yamamoto, Takashi Honda, Yosuke Inukai, Shinya Yokoyama, Takanori Ito, Norihiro Imai, Yoji Ishizu, Masanao Nakamura, Hiroki Kawashima

**Affiliations:** Department of Gastroenterology and Hepatology, Nagoya University Graduate School of Medicine, Nagoya 466-8560, Japanyokoshin@med.nagoya-u.ac.jp (S.Y.);

**Keywords:** chronic liver diseases, gut microbiome, prognostic factors, hepatocellular carcinoma, Barcelona Clinic Liver Cancer staging, 6S ribosomal RNA sequencing, Cox proportional hazards model, *Veillonella*, *Incertae sedis*, gut–liver axis

## Abstract

We investigated the prognostic role of the gut microbiome and clinical factors in chronic liver disease (hepatitis, cirrhosis, and hepatocellular carcinoma [HCC]). Utilizing data from 227 patients whose stool samples were collected over the prior 3 years and a Cox proportional hazards model, we integrated clinical attributes and microbiome composition based on 16S ribosomal RNA sequencing. HCC was the primary cause of mortality, with the Barcelona Clinic Liver Cancer staging system-derived B/C significantly increasing the mortality risk (hazard ratio [HR] = 8.060; 95% confidence interval [CI]: 3.6509–17.793; *p* < 0.001). Cholesterol levels < 140 mg/dL were associated with higher mortality rates (HR = 4.411; 95% CI: 2.0151–9.6555; *p* < 0.001). *Incertae sedis* from *Ruminococcaceae* showed a protective effect, reducing mortality risk (HR = 0.289; 95% CI: 0.1282 to 0.6538; *p* = 0.002), whereas increased *Veillonella* presence was associated with a higher risk (HR = 2.733; 95% CI: 1.1922–6.2664; *p* = 0.017). The potential of specific bacterial taxa as independent prognostic factors suggests that integrating microbiome data could improve the prognosis and treatment of chronic liver disease. These microbiome-derived markers have prognostic significance independently and in conjunction with clinical factors, suggesting their utility in improving a patient’s prognosis.

## 1. Introduction

The advent of next-generation sequencing (NGS) has revolutionized our understanding of the microbiome, revealing its intricate connections to a spectrum of diseases, including diabetes [1] and Parkinson’s disease [2]. This comprehensive analysis of bacterial populations has elucidated the pivotal role of the microbiome in health and disease, revealing how various components such as lipopolysaccharides (LPS), bile acids, and short-chain fatty acids directly influence liver inflammation through their flow into the liver via the portal vein, a connection underscored by the anatomical proximity of the liver and intestine [3].

Recent studies have highlighted the relationship between gut microbiota and diseases through new mechanisms, such as the link between primary sclerosing cholangitis and the gut microbiome [4]. Despite these advancements, the clinical application of NGS for microbiome analysis has yet to yield satisfactory results. Although the cost of such analyses has decreased, allowing for their widespread use, their clinical significance and practical applications still need to be fully established. Recent reports have discussed the potential of the NGS-based characterization of the gut microbiome in predicting short-term hospital outcomes [5] and the risk of complications such as hepatic encephalopathy [6] in cirrhosis.

In the present study, we attempted to identify bacteria associated with disease prognosis by analyzing stool samples collected more than three years ago. One of the primary challenges we faced was the intricate relationship between cirrhosis, hepatocellular carcinoma (HCC), and various clinical factors, all of which are complexly intertwined with the microbiome. Our approach involved separating these factors into clinical and microbiome categories for analysis and then integrating them to identify bacteria that were significantly associated with prognosis. This innovative method aims to bridge the gap between microbiome research and clinical applications by providing new insights into the prognostic potential of the gut’s microbiota composition.

## 2. Materials and Methods

### 2.1. Study Approval, Registration, and Patient Enrollment

This study was conducted with the approval of the Research Ethics Committee of Nagoya University Hospital, Japan (Protocol Number 2015-0420, dated 30 August 2016) and in strict adherence to the ethical principles of the Declaration of Helsinki. To protect patient privacy, all patient and sample data were anonymized and stored securely.

Between June 2016 and August 2019, 227 Japanese patients with chronic hepatitis or cirrhosis were enrolled in this study. Patients were carefully selected based on specified inclusion criteria, with the exclusion of recent antibiotic or immunosuppressive drug use within the preceding month, a history of gastric or colorectal surgery, or the presence of active infections.

### 2.2. Sample Collection and Microbial DNA Analysis

Sample collection and sequencing were performed as previously described [7]. Stool samples were obtained from participants under two distinct conditions: at home and in a hospital setting. Samples collected at home were immediately placed into a preservative solution designed to maintain microbial integrity, subsequently frozen at −80 °C within 48 h of collection. Differences between the two sampling methods have been reported to have no impact on the microbiome results [8,9]. Samples collected in the hospital were directly frozen at −80 °C to preserve microbial DNA.

For microbial DNA isolation, we used a DNeasy PowerSoil Kit (Qiagen, Hilden, Germany), which is known for its efficacy in extracting high-quality microbial DNA. DNA amplification targeted the V3–V4 regions of the bacterial 16S ribosomal RNA (rRNA) gene using universal primers.

Forward primer: 5′-TCGTCGGCAGCGTCAGATGTGTATAAGAGACAGCCTACGGGNGGCWGCAG-3′.

Reverseprimer: 5′-GTCTCGTGGGCTCGGAGATGTGTATAAGAGACAGGACTACHVGGGTATCTAATCC-3′.

Sequencing was performed using the Illumina MiSeq platform (Illumina, San Diego, CA, USA). Raw paired-end reads were processed using QIIME2 (version 2019.11) software [10], leveraging the DADA2 pipeline [11] for meticulous quality control, including filtering, trimming, denoising, alignment, and the taxonomic classification of Amplicon Sequence Variants (ASVs) against the Silva database [12].

### 2.3. Clinical Assessments

Comprehensive clinical data, including liver function tests and imaging studies (ultrasonography, CT, and MRI), were collected at baseline and every 3–6 months thereafter. HCC was diagnosed based on the imaging criteria and confirmed by at least two hepatologists. Questionnaires on lifestyle factors such as defecation frequency and exercise habits were also administered.

### 2.4. Statistical Analysis

In this study, Python programming language version 3.9 with statistics modules served as the foundation for conducting all statistical analyses and leveraging the extensive libraries known for statistical modeling and data analysis capabilities. Our methodology included the construction of Kaplan–Meier curves to graphically represent patient outcomes over time, complemented by log-rank test results to detect statistically significant differences in survival rates among distinct patient cohorts. Additionally, the Cox proportional hazards model was applied to the binarized microbiome data based on the median relative abundance of the bacterial taxa. This binary classification enables a nuanced evaluation of the effects of specific members of the microbiota on patient survival, highlighting the prognostic significance of these microbial compositions.

For the analysis of continuous variables, the Mann–Whitney U test was employed to compare distributions between two independent groups, ensuring the statistical integrity of our findings. Categorical data were analyzed using either the chi-square test or Fisher’s exact test, depending on the appropriateness of the dataset, to accurately determine the associations between the categorical variables and patient outcomes.

### 2.5. Demographic and Clinical Characteristics

The demographic variables analyzed included age, dichotomized into groups of under 65 and 65 or over, and sex. From a clinical perspective, we considered the etiology of liver disease and distinguished between viral and other causes. Other demographic attributes included the body mass index (BMI) with a cut-off value of 25, diabetes status denoting the presence or absence of the condition, and the Barcelona Clinic Liver Cancer (BCLC) staging system to differentiate between non-HCC cases, stages A or B, and stage C.

Medication use was comprehensively documented, including prescriptions for prednisolone (PSL), proton pump inhibitors (PPIs), probiotics, ursodeoxycholic acid (UDCA), and branched-chain amino acid (BCAA) supplements. Lifestyle factors were also rigorously evaluated, including smoking status (categorized as nonsmoker, ex-smoker, or current smoker), exercise habits (whether participants engaged in regular exercise), and dietary patterns, specifically focusing on the main diet type and vegetable intake. The presence of varices and stool characteristics (frequency and hardness) were examined in detail.

## 3. Results

### 3.1. Patient Background

This study included 227 patients diagnosed with chronic liver disease, with a predominantly male population (61.70%) and a median age of 68 years (Table 1). The median survival time after sampling was 721 days. Considering etiology, the hepatitis C virus (HCV) was the most common etiology, affecting 57.70% of the cohort, followed by the hepatitis B virus (HBV) (20.30%), non-HBV/non-HCV (NBNC) liver disease (11.50%), and alcohol-related liver disease (10.60%). Most participants were classified as Child–Pugh Class A (84.60%), denoting a milder severity of liver impairment in our patient cohort. BCLC staging indicated that 25.20% of patients were at stage A. Diabetes was prevalent in 30% of the cohort, and nearly half (48%) were identified as having active HCC. Varices were observed in 27.30% of patients.

Lifestyle assessments revealed 59.40% were nonsmokers and 70.50% were abstainers from habitual alcohol consumption. Dietary habits showed a preference for fish (53.80%) and adequate vegetable consumption (74.50%), with a median stool frequency of seven times per week. Regarding medication, PPIs were used by 29.50% of the patients, UDCA by 33.90%, and BCAA supplements by 14.50%.

Laboratory findings revealed a median total protein level of 7.2 g/dL, an albumin level of 3.9 g/dL, and aspartate aminotransferase (AST) and alanine aminotransferase (ALT) median levels of 32 and 28 U/L, respectively. Renal function tests showed a median creatinine level of 0.76 mg/dL and an estimated glomerular filtration rate (GFR) of 71.8 mL/min/1.73 m^2^. The median value of the inflammatory marker, the C-reactive protein (CRP), presented a median value of 0.1 mg/L. Hematological assessments reported median hemoglobin levels of 13.3 g/dL and platelet counts of 155 × 10^3^ cells/µL. The median albumin–bilirubin index (ALBI) was −2.62, with a Fibrosis-4 index of 2.89 and a Prognostic Nutritional Index of 47, reflecting the nutritional and fibrotic status of this patient cohort.

### 3.2. Survival Curves and Causes of Death for All Patients

The Kaplan–Meier survival analysis delineated the temporal progression of patient survival, with the median survival time exceeding the study duration, indicating prolonged survival for a significant segment of the cohort (Figure 1). Survival probabilities were estimated to be 93.3% after one year and 76.6% after three years, demonstrating a relatively stable survival rate in the initial years post-sampling.

A half-yearly assessment of mortality causes for up to 3 years highlighted a gradual increase in deaths attributable to HCC that became progressively more evident. By the end of the first year, the influence of HCC on mortality rates became significantly apparent, with an ascending trajectory of HCC-related deaths noted at subsequent intervals (Figure 2).

At the end of the study period, HCC was the principal cause of death, accounting for 65.7% of all deaths. Liver cirrhosis was responsible for 22.9% of deaths, with the remaining 11.4% of deaths attributed to various causes, including esophageal varices, pneumonia, lung cancer, peritonitis, and cerebral hemorrhage. This distribution underscores the predominance of HCC as a critical mortality factor in this patient population along with the diverse array of complications that contribute to the overall mortality rate.

### 3.3. Clinical Factors Associated with Prognosis

The univariate Cox regression analysis identified a spectrum of significant predictors affecting the prognosis of patients with chronic liver disease (Figure 3, Appendix A). The comprehensive analysis included demographic variables, clinical characteristics, laboratory values, and lifestyle factors. Notable findings included the association of an ALBI grade greater than −2.6 with a hazard ratio (HR) of 4.673 (95% CI: 2.0413–10.7000, *p* < 0.001), indicating a significantly increased risk. Similarly, alkaline phosphatase levels exceeding 322 U/L, habitual or heavy alcohol consumption, the non-use of BCAA supplements, hemoglobin levels below 12 g/dL, lactate dehydrogenase levels above 222 U/L, and platelet counts below 150 × 10^9^/L were found to significantly affect survival outcomes. Additional factors such as sex, with women exhibiting a reduced risk (HR: 0.422, 95% CI: 0.1916–0.9296, *p* = 0.032), and sodium levels below 140 mmol/L were also identified as significant. Notably, total cholesterol levels below 140 mg/dL emerged as a high-risk factor, with an HR of 4.627 (95% CI: 2.3810–8.9933, *p* < 0.001).

After adjusting for confounders in the multivariable analysis, the three predictors mentioned above remained statistically significant, underscoring their robust impact on patient prognosis (Figure 4, Appendix A). Advanced BCLC stage B/C was the most potent predictor, with an adjusted HR of 6.715 (95% confidence interval [CI]: 3.7336–16.0499; *p* < 0.001). The female sex was associated with a lower risk, marked by an adjusted HR of 0.328 (95% CI: 0.1916–0.9296; *p* = 0.043). The analysis also confirmed that a total cholesterol level of <140 mg/dL significantly increased the risk, with an adjusted HR of 2.867 (95% CI: 2.3810–8.9933, *p* = 0.015). This multivariable approach allows for a nuanced understanding of the contribution of each factor to prognosis by factoring in the complex interplay between multiple variables.

### 3.4. Factors of Bacteria Associated with Prognosis

The analysis of the binarized gut microbiome data, as detailed in Figure 5 and Appendix A, utilized a univariate Cox proportional hazards framework to assess the 47 bacterial genera identified with median relative abundances above zero. This approach segmented patients into groups according to whether the abundance of each bacterial genus exceeded or fell below the median. The resulting forest plot illustrates the log HRs, delineating the survival implications of these bacterial taxa in chronic liver disease. Notably, an above-median abundance of Incertae sediments from the *Ruminococcaceae* family was associated with a diminished risk (HR: 0.3989), whereas an increased abundance of *Streptococcus* was associated with an increased risk (HR: 3.6655). Additional genera such as *Lactobacillus*, *Oscillibacter*, and *Veillonella* were also represented, highlighting their risk associations contingent on relative abundance.

Building on this foundation, Figure 6 and Appendix A present the findings of the multivariable Cox regression analysis employing the same binarized bacterial genera dataset. This analysis, which was adjusted for potential confounders, continued to explore the influence of bacterial abundance on patient outcomes. The protective effect observed with *Incertae sedis* from the *Ruminococcaceae* family in the univariate analysis (HR: 0.3989) was corroborated by the multivariate analysis, as reflected in an adjusted HR of 0.4319. Similarly, *Streptococcus* maintained its status as a significant risk factor (adjusted HR: 2.7321), underscoring the persistent effects of bacterial abundance on survival prospects within this patient cohort.

### 3.5. Integrated Analysis of Clinical and Microbiome Predictors of Survival

In this comprehensive multivariable analysis, we examined the prognostic significance of both clinical parameters and microbiome constituents on the survival outcomes of patients with chronic liver disease (Figure 7, Appendix A). Critical factors, such as the BCLC stage and total cholesterol levels, were evaluated along with specific microbiota to determine their impact on prognosis. An advanced BCLC stage, categorized as non-HCC or BCLC stage A versus BCLC stages B and C, was identified as a significant predictor of an 8-fold increase in mortality risk (HR = 8.06, coef = 2.087, *p* < 0.0001). Additionally, cholesterol levels at or below 140 mg/dL were linked to a 4.4 times higher risk of mortality (HR = 4.411, coef = 1.484, *p* = 0.0002), highlighting the adverse prognostic impact of lower cholesterol levels.

The microbiome analysis further revealed that the relative abundance of *Incertae sedis* from the *Ruminococcaceae* family (an above-median presence) was associated with a 71% reduction in mortality risk (HR = 0.29, coef = −1.24, *p* = 0.003), indicating a protective effect. By contrast, a higher median presence of *Veillonella* significantly increased mortality risk by nearly 2.7-fold (HR = 2.733, coef = 1.005, *p* = 0.018), indicating that it is a noteworthy prognostic factor.

### 3.6. Overall Survival Analysis by Microbiome Relative Abundance

Kaplan–Meier survival curves were constructed for all patients and classified into two groups based on the median relative abundance of each bacterial genera. The analysis revealed that patients with an above-median presence of *Incertae sedis* from the *Ruminococcaceae* family had a significantly lower risk of mortality (*p* = 0.009; Figure 8), indicating a potential protective effect on patient survival. Conversely, an above-median relative abundance of *Veillonella* was associated with an increased mortality risk (*p* = 0.005; Figure 9), suggesting an adverse impact on survival outcomes.

### 3.7. BCLC Stage-Specific Survival Impact

Kaplan–Meier survival analyses were conducted for patients categorized by BCLC staging into BCLC group 0 (non-HCC or BCLC stage A) and group 1 (BCLC stages B and C). The survival analysis for patients in BCLC group 0 showed a statistically significant difference for *Incertae sedis* from the *Ruminococcaceae* family, with a log-rank *p*-value of 0.04238 (Figure 10), indicating a notable difference in survival between patients with high (n = 78) and low (n = 79) abundances of this microbial genera. However, for *Veillonella*, the difference in survival within BCLC group 0 was not statistically significant (*p* = 0.1784, Figure 11).

In BCLC Group 1, both *Incertae sedis* from the *Ruminococcaceae* family and *Veillonella* showed statistically significant differences in survival. The log-rank *p*-value was 0.042 (Figure 12) for *Incertae sedis* from the *Ruminococcaceae* family and 0.02184 (Figure 13) for *Veillonella*, with high-abundance groups (n = 30 for *Incertae sedis* from the *Ruminococcaceae* family and n = 31 for *Veillonella*) showing varying survival curves compared to the low-abundance groups (n = 29 for *Incertae sedis* from the *Ruminococcaceae* family and n = 28 for *Veillonella*).

## 4. Discussion

Chronic liver disease presents a significant public health challenge, with cirrhosis and HCC being the predominant causes of mortality in this patient group. Our study underscores the critical role of the liver cancer stage, particularly the BCLC B/C classification, in influencing patient outcomes and overshadowing factors such as cirrhosis severity and age. Despite the vast array of identified bacterial genera in the gut microbiome, our analysis identified a select group of 47 genera that were consistently associated with patient prognosis. Notably, the genus *Veillonella* and a member of the *Ruminococcaceae* family, *Incertae sedis*, have emerged as key prognostic markers and continue to be significantly correlated with patient outcomes across diverse analytical models. Their abundance or scarcity is directly linked to patient survival prospects, highlighting their potential as biomarkers for disease progression and treatment response.

Chronic liver disease is a major global public health concern. Cirrhosis and HCC are the primary causes of death in patients with chronic liver conditions [13]. Cirrhosis is the most common precursor of HCC and accelerates its progression to liver cancer. Cirrhosis and HCC share closely related pathological processes, and cirrhosis is widely recognized [14] as the leading pathway for the development of liver cancer. Liver function plays a critical role in selecting a treatment for HCC; however, patients with severe cirrhosis have limited treatment options, often leading to a poorer prognosis. Moreover, the treatment of HCC can accelerate the progression of cirrhosis in some cases.

The identification of prognostic factors is of utmost importance as it provides essential guidance for the management and selection of treatment strategies for patients. Traditional studies have reported the use of indices for cirrhosis, such as the Child–Pugh classification and ALBI Score [15], and for HCC, the BCLC stage classification [16]. Generally, HCC progresses faster than cirrhosis. It is also known that women with chronic liver disease tend to have a better prognosis than men. Consequently, in the present study, the relatively short observation period caused HCC to emerge as the most significant cause of death. Additionally, the identification of a low cholesterol level as a prognostic factor for worsening outcomes is an important indicator of metabolic abnormalities in chronic liver disease. The liver plays a central role in the cholesterol metabolism; therefore, liver dysfunction often leads to abnormalities in the cholesterol metabolism, which may be associated with worse outcomes in patients with chronic liver disease [17].

The relationship between the microbiome and liver is complex. The intestinal barrier and liver physiology underscore a bidirectional relationship in which both organ systems significantly influence each other’s functional and health status. This relationship has been increasingly recognized for its critical role in the pathogenesis and progression of various liver diseases, including non-alcoholic fatty liver disease, alcoholic liver disease, hepatitis, and cirrhosis [18,19]. The composition of the gut microbiome and its metabolic products can profoundly affect liver health through multiple mechanisms, including the modulation of immune responses, the production of toxic metabolites, and their impact on metabolic homeostasis.

The gut–liver axis refers to the reciprocal interactions between the gut, its microbiota, and the liver. These include the direct transport of gut-derived substances to the liver through the portal vein, the impact of the liver on bile acid composition and gut microbiota through bile secretion, and systemic immune interactions between the gut and liver. The integrity of the intestinal barrier, the compositional balance of the gut microbiota, and immune system functions are pivotal for maintaining the health of this axis. The disruption of any of these components can lead to or exacerbate the liver disease.

In this study, we identified members of *Incertae sedis* within the *Ruminococcaceae* family, a crucial but taxonomically unresolved group within the human gut microbiome. These bacteria are integral to fermenting dietary fibers and facilitate the production of short-chain fatty acids (SCFAs), which are essential for maintaining gut health and systemic well-being. Despite their indeterminate classification, these organisms are vital for the digestion of complex carbohydrates, the enhancement of energy availability, and the positive modulation of the gut environment [20]. The diversity within the *Ruminococcaceae* family contributes significantly to the integrity of the gut barrier, the modulation of the immune system, and protection against pathogens [21]. The production of SCFAs by these bacteria is particularly noteworthy given their association with a range of health benefits, including anti-inflammatory properties, protection against colorectal cancer, and the regulation of blood glucose and cholesterol levels. Additionally, the identification of *Veillonellaceae* and *Streptococcus* in the gut microbiome directed attention to the oral–gut axis, highlighting the movement of oral microbes to the gut and their potential impact on gastrointestinal and systemic health.

Recent investigations have revealed significant correlations between the gut microbiome and liver diseases, particularly cirrhosis, suggesting potential avenues for therapeutic interventions. A foundational study by Bajaj et al. [22] delineated the altered profiles of the gut microbiome across varying severities of cirrhosis and introduced the cirrhosis dysbiosis ratio (CDR) as a measure of microbial imbalance. This study highlighted a decrease in beneficial bacteria such as Clostridiales XIV, *Lachnospiraceae*, and *Ruminococcaceae,* along with an increase in potentially pathogenic taxa such as *Enterobacteriaceae* and *Veillonella*, which correlated with disease progression. In the Japanese population, the community type of the gut microbiome is divided into five main categories. Liver disease is highly prevalent in Type A and D patients. The relative abundance of *Ruminococcus*, *Streptococcus*, and *Veillonella* was higher than that of Type E, which was dominated by healthy individuals [23].

Following this line of inquiry, Bajaj et al. subsequently [5] focused on the predictive value of microbiota for clinical outcomes in hospitalized patients with cirrhosis. These findings underscore the prognostic significance of alterations in beta diversity, and the association of specific microbial communities with an increased risk of organ failure, acute-on-chronic liver failure, and mortality. This underscores the potential of gut dysbiosis as a prognostic biomarker for liver cirrhosis.

We also explored the specific influence of oral bacteria on liver health and identified a correlation between the abundance of *Streptococcus* and *Veillonella* and ALBI grade in patients with hepatitis C [24]. This association is particularly relevant given the reported link between certain bacteria, including *Veillonella*, and the heightened risk of hepatic encephalopathy, a severe complication of liver disease. Thus, the presence of these bacteria in the gut microbiome may serve as a potential indicator of disease severity and the risk of complications such as hepatic encephalopathy, emphasizing the need for comprehensive microbial profiling in clinical assessments [6].

Wu et al. [25] advanced this field by demonstrating the efficacy of adjuvant probiotic therapy in modifying the gut microbiota to improve outcomes in liver cirrhosis. Their study highlighted the critical role of SCFA-producing bacteria, which were significantly reduced in cirrhosis, particularly during the decompensatory stages. The introduction of probiotics, aimed at enhancing the abundance of these beneficial microbes, has led to biochemical improvements, suggesting a viable therapeutic strategy to counteract the effects of microbial dysbiosis in cirrhosis.

By incorporating our findings, this comprehensive analysis underscores the significant correlation between the gut microbiome and liver disease progression, particularly the shift toward pathogenic profiles in advanced cirrhosis and their link to increased risks such as hepatic encephalopathy. Our study reinforces the notion that microbiome constituents, specifically *Veillonella* and *Incertae sedis* from the *Ruminococcaceae* family, could serve as independent prognostic markers, offering new insights into the role of the gut–liver axis in disease progression. The integration of microbiome profiling with conventional clinical markers, particularly liver cancer staging, has emerged as a promising approach to enhance the predictive accuracy of patient outcomes in chronic liver disease. These findings provide a more nuanced understanding of microbial influences and support the development of personalized, microbiome-informed therapeutic strategies, thereby opening new avenues for targeted interventions aimed at modifying gut microbiota to improve liver health and patient prognosis.

## 5. Conclusions

Over a 721-day study period, HCC was identified as the primary cause of mortality and BCLC B/C staging emerged as a key prognostic indicator. Remarkably, low levels of *Incertae sedis* from the *Ruminococcaceae* family, alongside high levels of *Veillonella*, were consistently linked with poor patient prognosis, underscoring their potential as unique prognostic markers for chronic liver disease.

## Figures and Tables

**Figure 1 microorganisms-12-00610-f001:**
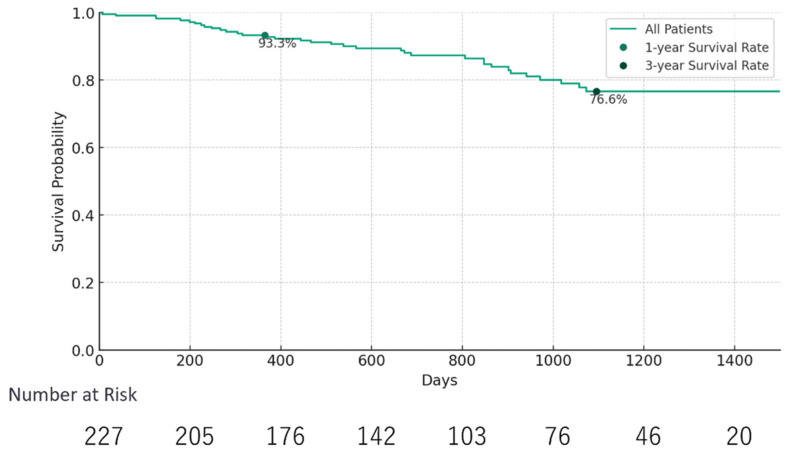
Kaplan–Meier survival curves for patients with chronic liver disease. The Kaplan–Meier survival curves depicting the survival probabilities of patients with chronic liver disease over time are shown. The graph highlights the survival rates at various checkpoints, including those at 1, 3, and 5 years, providing a visual representation of the longitudinal survival trends within the patient cohort. The curve underscores the high survival probability within the first year post-sampling, which gradually declines over subsequent years. Statistical significance was assessed using the log-rank test, with a focus on comparing different patient subgroups based on their microbiome composition and other clinical factors.

**Figure 2 microorganisms-12-00610-f002:**
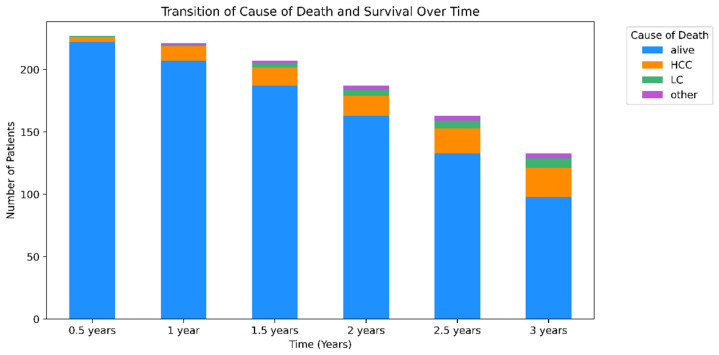
Distribution of causes of death over time. The bar graph presents the distribution of the causes of death among patients with chronic liver disease at half-year intervals for up to 3 years. Each bar shows the proportion of deaths attributed to hepatocellular carcinoma (HCC), liver cirrhosis (LC), or other causes, with the remaining portion representing patients who remained alive. The analysis revealed that 65.7% of all deaths were due to HCC, 22.9% were due to LC, and 11.4% were due to other causes, including esophageal varices, pneumonia, lung cancer, peritonitis, and cerebral hemorrhage. This visual breakdown provides insights into the changing landscape of the causes of mortality over time, highlighting the predominant role of HCC as a leading cause of death in this patient population.

**Figure 3 microorganisms-12-00610-f003:**
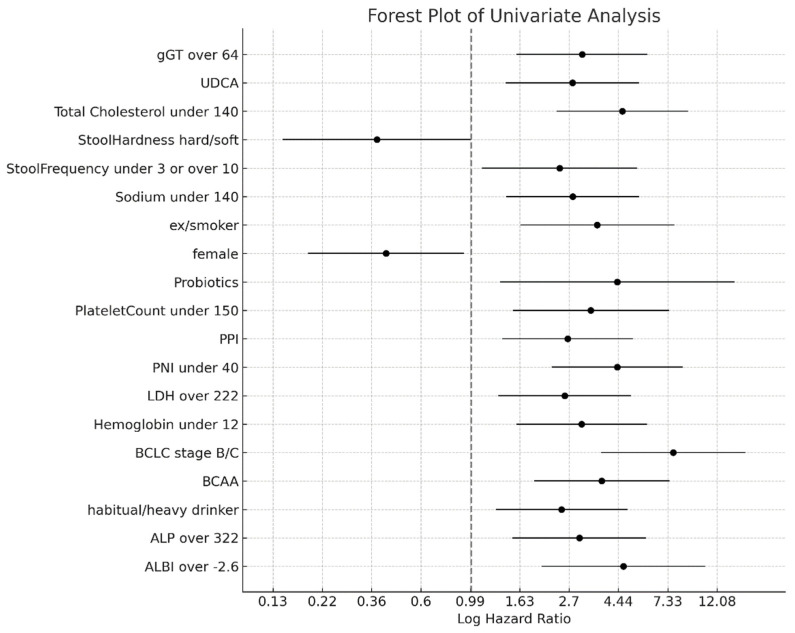
Forest plot of significant predictors from the univariate cox regression analysis. This forest plot delineates the outcomes of the univariate Cox regression analysis by showing predictors that significantly influence the prognosis of patients with chronic liver disease. Each data point symbolizes the log-transformed hazard ratio (HR) as a key predictor, accompanied by horizontal lines representing 95% confidence intervals (CIs). The analysis encompassed a broad range of factors, including demographic details, clinical features, laboratory measurements, and lifestyle habits such as age, sex, etiology, body mass index (BMI), diabetes status, the Barcelona Clinic Liver Cancer staging, medication intake (prednisolone: PSL, proton pump inhibitors: PPIs, probiotics, ursodeoxycholic acid: UDCA, and the branched-chain amino acid: BCAA), the presence of varices, Prognostic Nutritional Index (PNI), estimated glomerular filtration rate (eGFR), platelet count, alcohol consumption patterns, and total cholesterol. The vertical axis intersecting the plot symbolizes the no-effect threshold (HR of 1), with data points situated to the right indicating an elevated risk, and those to the left indicating a reduced risk associated with each predictor. Dotted line shows hazard ratio 1.

**Figure 4 microorganisms-12-00610-f004:**
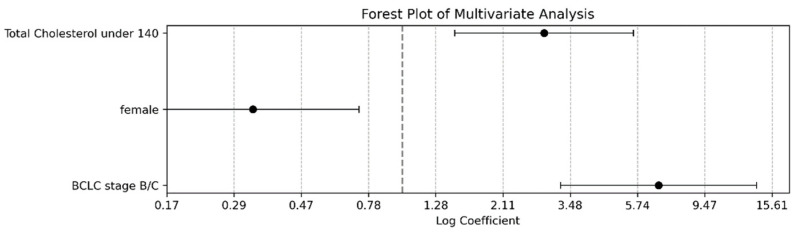
Adjusted predictor impact on patient outcomes: multivariate analysis forest plot. This forest plot illustrates the adjusted effects of specific predictors on patient outcomes in the context of chronic liver disease, accounting for potential confounders through a multivariate analysis. We focused on statistically significant factors after adjustment, including their BCLC stage, sex, and total cholesterol levels. Each log-transformed HR is depicted alongside its 95% CIs, mirroring the format of the univariate analysis plot. This multivariate approach evaluated the combined influence of all variables within the model, offering a comprehensive insight into the distinct impact of each predictor on patient prognosis. The plot serves as a crucial tool for identifying the most substantial predictors to guide clinical decision-making and optimize patient care strategies.

**Figure 5 microorganisms-12-00610-f005:**
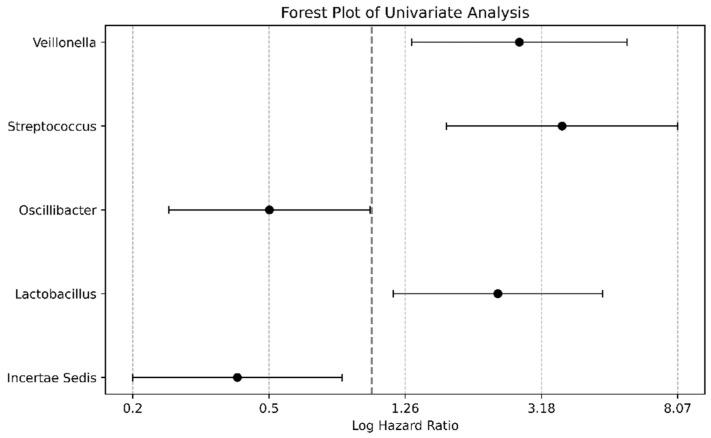
Univariate analysis forest plot with binarized bacterial data. The forest plot was derived from the univariate Cox proportional hazards analysis, which incorporated 47 identified bacterial genera with median relative abundances above zero. For this analysis, the patients were divided into two groups based on whether the abundance of each bacterial genus was above or below the median value. The plot shows the log hazard ratios (HRs), indicating a relationship between the presence of these bacterial taxa and patient survival in chronic liver diseases. The genus *Incertae sedis* from the *Ruminococcaceae* family, with an abundance above the median, was associated with a decreased risk (HR: 0.3989), in contrast to *Streptococcus*, whose higher abundance correlated with an increased risk (HR: 3.6655). The genera *Lactobacillus*, *Oscillibacter*, and *Veillonella* are also plotted, showing their respective risk associations based on their abundances relative to the median.

**Figure 6 microorganisms-12-00610-f006:**
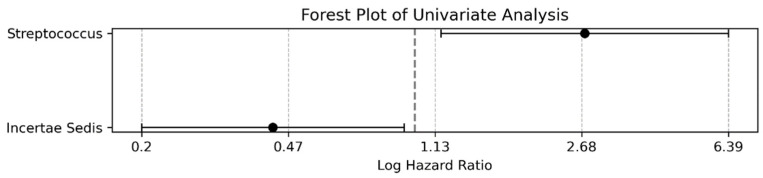
Multivariate analysis forest plot with binarized bacterial data. This shows a forest plot for a multivariate Cox regression analysis. It includes the same bacterial genera as in the univariate analysis that have been binarized around their median values. This multivariate approach was adjusted for confounders when evaluating the impact of bacterial abundance on patient outcomes. The protective effect of *Incertae sedis* from the *Ruminococcaceae* family persisted in this analysis (adjusted HR: 0.4319) and *Streptococcus* remained a significant risk factor (adjusted HR: 2.7321), reinforcing the implications of bacterial abundance on the survival of chronic liver disease.

**Figure 7 microorganisms-12-00610-f007:**
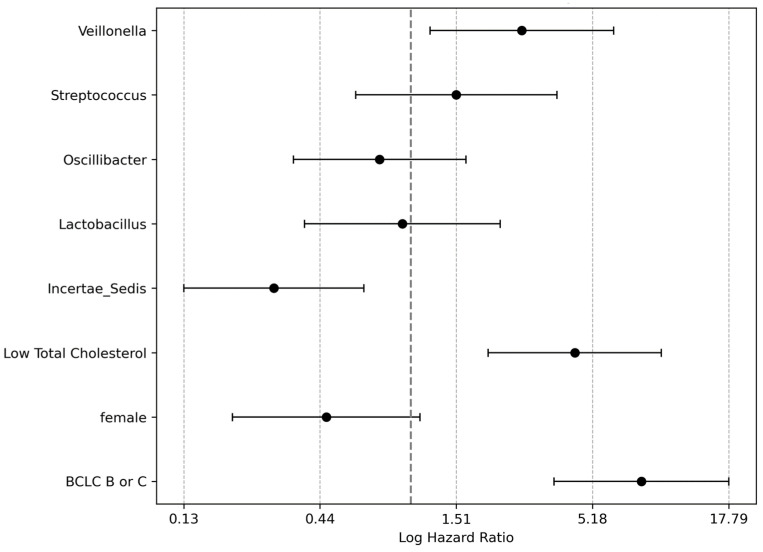
Multivariable analysis forest plot of chronic liver disease patients. This forest plot displays the outcomes of the multivariate analysis, illustrating the adjusted hazard ratios (HRs) for significant predictors of prognosis in patients with chronic liver disease. This analysis highlights the impact of the presence of specific bacteria, namely *Incertae sedis* from the *Ruminococcaceae* family and *Veillonella*, on patient survival, along with other clinical parameters. Each point represents the log-transformed HR as a significant predictor, with the horizontal lines indicating 95% confidence intervals (CIs). Notably, the plot underscores the significant role of certain microbiome compositions adjusted for clinical factors, such as Barcelona Clinic Liver Cancer staging stage, sex, and total cholesterol levels. This visualization aids in identifying predictors that significantly influence the survival of patients with chronic liver disease, guiding therapeutic strategies and patient care.

**Figure 8 microorganisms-12-00610-f008:**
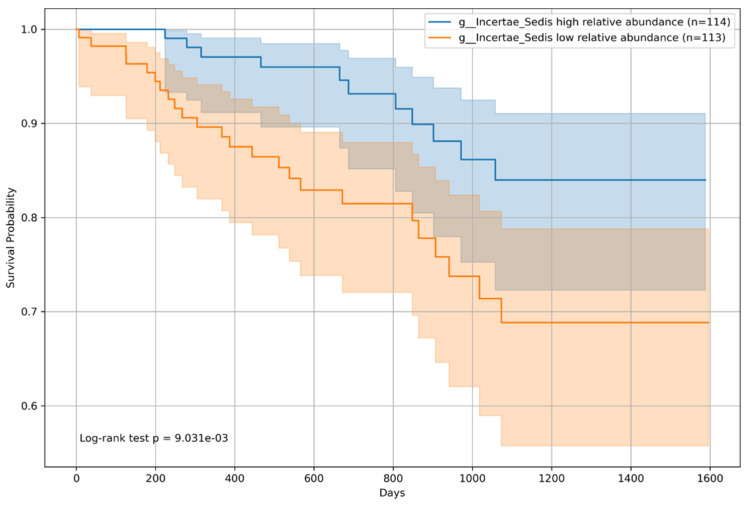
Kaplan–Meier survival curve for *Incertae sedis* from the *Ruminococcaceae* family. This figure shows Kaplan–Meier survival curves comparing the survival probabilities of patients with chronic liver disease with a high versus a low relative abundance of *Incertae sedis* from the *Ruminococcaceae* family. The curves demonstrate distinct survival trajectories over time, with the statistical analysis conducted using the log-rank test (*p* = 9.031 × 10^−3^). The analysis revealed a significant difference in survival rates between patients with high (n = 114) and low (n = 113) relative abundances of *Incertae sedis* from the *Ruminococcaceae* family, suggesting that this bacterial genus influences patient prognosis. The inclusion of patient counts in the legend and log-rank test results in the figure facilitate a comprehensive understanding of the data.

**Figure 9 microorganisms-12-00610-f009:**
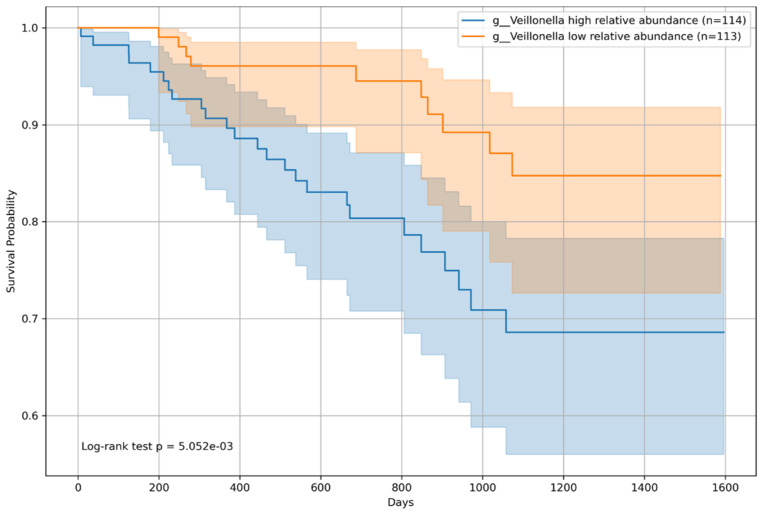
Kaplan–Meier survival curve for *Veillonella*. The Kaplan–Meier survival curves for patients categorized according to the relative abundance of *Veillonella* are shown. Survival analysis indicated a significant difference (*p* = 5.052 × 10^−3^) between the survival probabilities of patients with high (n = 114) and low (n = 113) abundances of *Veillonella*. The figure illustrates the impact of *Veillonella* on the survival of patients with chronic liver disease, supported by the log-rank test. Detailed annotations, including patient group sizes and *p*-values, enrich the interpretability of the survival trends depicted in the figure.

**Figure 10 microorganisms-12-00610-f010:**
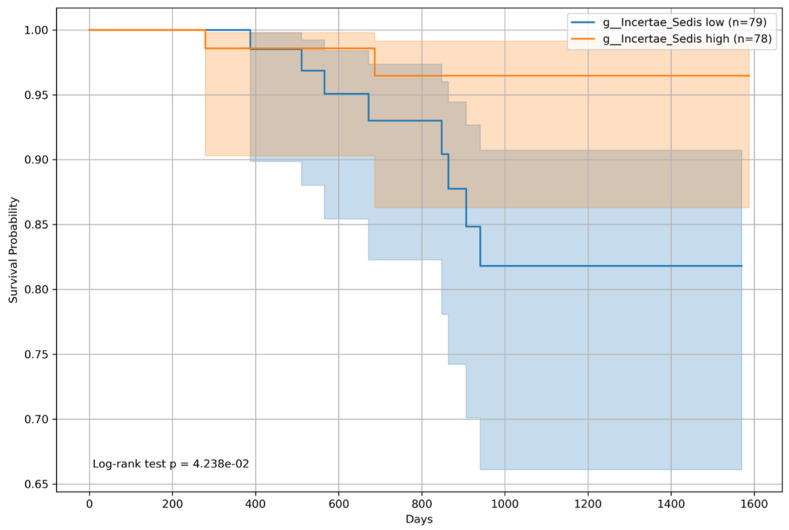
Kaplan–Meier survival curve for patients with *Incertae sedis* from the *Ruminococcaceae* family and the Barcelona Clinic Liver Cancer staging (BCLC) group 0 (non-HCC and Stage A). The survival probabilities for patients with BCLC 0 are depicted and differentiated by the median relative abundance of Incertae_Sedis. The log-rank test revealed a significant difference in survival, favoring the high-abundance group (*p* = 0.04).

**Figure 11 microorganisms-12-00610-f011:**
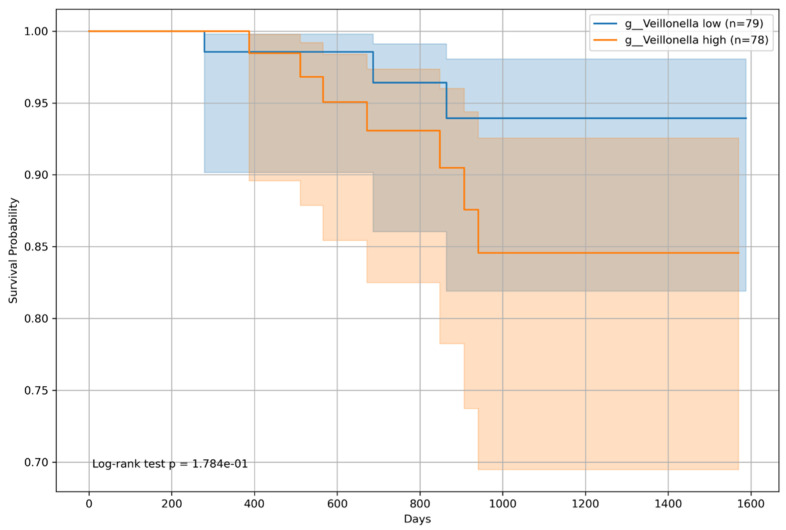
Kaplan–Meier survival curve for patients with *Veillonella* and the Barcelona Clinic Liver Cancer staging (BCLC) group 0. This represents the survival probabilities of patients with BCLC 0 stratified by the median relative abundance of *Veillonella*. The log-rank test indicated no significant difference in survival between the high- (n = 78) and low-abundance (n = 79) groups (*p* = 0.1784).

**Figure 12 microorganisms-12-00610-f012:**
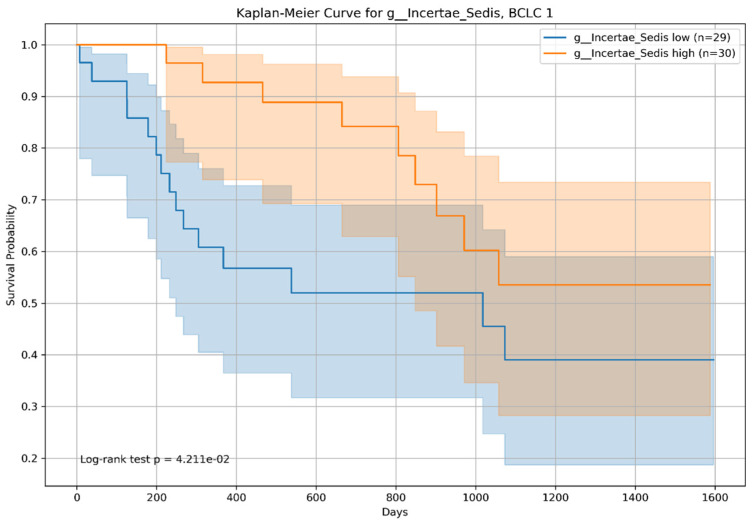
Kaplan–Meier survival curve for patients with *Incertae sedis* from the *Ruminococcaceae* family and the Barcelona Clinic Liver Cancer staging (BCLC) group 1 (BCLC Stages B and C). This illustrates the survival probabilities of patients with BCLC stage 1 stratified by the median relative abundance of Incertae_sedis. The log-rank test confirmed a significant difference in survival between the high- and low-abundance groups (*p* = 0.04211).

**Figure 13 microorganisms-12-00610-f013:**
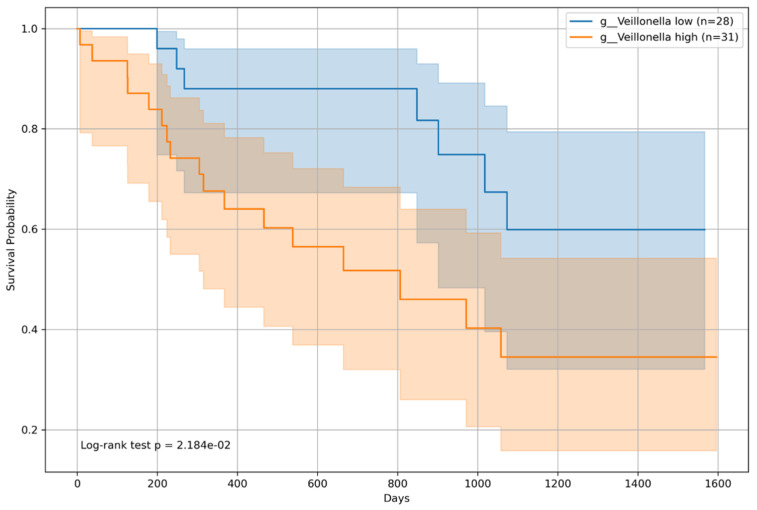
Kaplan–Meier survival curve for patients with *Veillonella* and the Barcelona Clinic Liver Cancer staging (BCLC) group 1. This shows the survival probabilities of patients classified as having BCLC 1 segregated by the median relative abundance of *Veillonella*. A significant difference in survival was observed, with the log-rank test yielding a *p* value of 0.02184, highlighting the impact of *Veillonella* on patient prognosis.

**Table 1 microorganisms-12-00610-t001:** Patient background.

Age	Years	68	(55–74)
Sex	Female	87	38.30%
	Male	140	61.70%
Survival Time	Days	721	(453–1125.5)
Etiology	Alcohol	24	10.60%
	HBV	46	20.30%
	HCV	131	57.70%
	NBNC	26	11.50%
Child–Pugh Class	A	192	84.60%
	B	31	13.70%
	C	4	1.80%
BCLC	A	55	25.20%
	B	43	19.70%
	C	16	7.30%
	No HCC	104	47.70%
Diabetes	Yes	68	30%
	No	159	70%
Varices	Yes	62	27.30%
	No	69	30.40%
	Unknown	96	42.30%
**Lifestyle**			
Smoking	Ex-smoker	43	26.10%
	Nonsmoker	98	59.40%
	Smoker	24	14.50%
Alcohol Consumption	Habitual	34	15.20%
	Heavy	32	14.30%
	Non	158	70.50%
Exercise	Yes	72	45.90%
	No	85	54.10%
Main Diet	Fish	85	53.80%
	Meat	73	46.20%
Vegetables	Enough	120	74.50%
	Not enough	41	25.50%
Stool Hardness	Hard	5	2.40%
	Normal	16	7.70%
	Soft	188	90%
Stool Frequency	Times/day	7	(5–7)
Medication			
Proton Pump Inhibitors	Yes	67	29.50%
	No	160	70.50%
Probiotics	Yes	7	3.10%
	No	220	96.90%
Prednisolone	Yes	4	1.80%
	No	223	98.20%
Ursodeoxycholic acid	Yes	77	33.90%
	No	150	66.10%
Laxative	Yes	31	13.70%
	No	196	86.30%
Branched-Chain Amino Acid	Yes	33	14.50%
	No	194	85.50%
Blood Test			
Total Protein	g/dL	7.2	(6.9–7.5)
Albumin	g/dL	3.9	(3.5–4.3)
Aspartate Aminotransferase	U/L	32	(22–50.5)
Alanine Aminotransferase	U/L	28	(17.5–48)
Alkaline Phosphatase	U/L	271	(221.5–377.5)
Gamma-Glutamyl Transferase	U/L	36	(23.5–73.5)
Cholinesterase	U/L	270	(188–323.5)
Total Bilirubin	mg/dL	0.8	(0.6–1.2)
Direct Bilirubin	mg/dL	0.1	(0.1–0.2)
Lactate Dehydrogenase	U/L	198	(174.5–233.5)
Amylase	U/L	88	(71.25–113)
Blood Urea Nitrogen	mg/dL	14.95	(12.6–18.1)
Creatinine	mg/dL	0.76	(0.61–0.92)
Estimated Glomerular Filtration Rate	mL/min/1.73 m^3^	71.8	(61.35–86.02)
Uric Acid	mg/dL	5.4	(4.3–6.5)
Osmolarity	mOsm/kg	282	(278.25–284)
White Blood Cells	cells/μL	4.8	(3.9–5.85)
Red Blood Cells	cells/μL	4.3	(3.92–4.72)
Hematocrit	%	39.7	(36.2–42.95)
Hemoglobin	g/dL	13.3	(12.1–14.65)
Platelet Count	cells/μL	155	(106.5–207.5)
Sodium	mmol/L	141	(139–142)
Potassium	mmol/L	4.2	(4–4.4)
Total Calcium	mg/dL	4.8	(4.5–9.2)
Magnesium	mEq/L	1.7	(1.6–2)
Prothrombin Time	%	95.8	(84.05–103.45)
C-Reactive Protein	mg/L	0.1	(0.05–0.27)
Glucose	mg/dL	102	(91–123)
Total Cholesterol	mg/dL	175	(148–195.5)
Albumin-Bilirubin Index		−2.62	(−2.86–2.19)
Fibrosis-4 Index		2.89	(1.66–5.83)
Prognostic Nutritional Index		47	(41.5–50.5)

BCLC, Barcelona Clinic Liver Cancer.

## Data Availability

Data supporting the findings of this study are available from the Department of Gastroenterology and Hepatology at the Nagoya University Graduate School of Medicine, Japan. However, restrictions apply to the availability of these data, which were used under a license for the current study, and, thus, are not publicly available. The data are available from the corresponding author upon reasonable request and with permission from the Department of Gastroenterology and Hepatology, Nagoya University Graduate School of Medicine, Japan.

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
