# Peer review of "Identification of the Microbiome Associated with Prognosis in Patients with Chronic Liver Disease"

_microorganisms, 2024, doi:10.3390/microorganisms12030610_

Round 1

Reviewer 1 Report

Comments and Suggestions for Authors

Dear authors,

We have carefully reviewed your manuscript and have several suggestions for improvement:

  1. The abstract lacks quantitative results and a clear conclusion. We kindly ask you to revise it to include these elements.

  2. The quality of the figures is suboptimal. We recommend improving them to enhance readability and clarity for the readers.

  3. It would be beneficial to provide more detailed information about the multivariate analysis. Additionally, we suggest organizing all figures depicting hazard ratios together to facilitate comparison.

  4. In the first paragraph of the discussion section, please provide a concise summary of the key results you intend to discuss, followed by a detailed analysis of each point.

  5. The conclusion section is overly lengthy and contains an excessive amount of results. We advise trimming it down to succinctly present your main conclusions.

  6. Following the aforementioned revisions, please ensure that the abstract aligns with the updated content of the manuscript.

Thank you for your attention to these matters. We look forward to receiving the revised manuscript.

Sincerely,

Reviewer 2 Report

Comments and Suggestions for Authors

The manuscript with the title "Identification of the microbiome associated with the prognosis in patients with chronic liver disease" presents the important, essential results of a 3-year study, in order to establish the most accurate prognosis of chronic liver disease, as well as the application of alternative therapies and not only, personalized.

The manuscript is well written, very interesting from a scientific point of view, with practical applicability, as well as a standard model of analyses. I would have some recommendations:

- please write all the names of bacteria in Latin, in italics.

- Regarding the most significant bacteria: Veillonella and Incertae sedis, if you have data on their levels before the onset or confirmation of the disease?

-Could supplements with these bacteria constitute treatments?

Round 2

Reviewer 1 Report

Comments and Suggestions for Authors

Dear Authors:

Thank you for revising the paper, please:

1- the figures still have some problem, there are different parts in Fig 2 and 3, why? please provide single Figure or each figure number,

2-You cannot name a Figure 4a! Please number them accordingly (Figure 4, 5, 6, ...) if you put all in one Figure you can do such a thing.

Author Response

Dear Reviewer,

Thank you very much for your valuable feedback on our manuscript. In response to your comments, we have carefully revised the figures as suggested.

We have renumbered all the figures in the manuscript. The revised figure numbers have been highlighted with a blue background color to easily distinguish the updates. This adjustment ensures each figure is uniquely and sequentially numbered, as you recommended.

Additionally, we have reinstated the figure descriptions from the initially reviewed manuscript, as the modifications necessitated further clarifications. We believe that these enhancements have made the figures more coherent and the manuscript as a whole more comprehensible.

We appreciate your thorough review and constructive suggestions, which have significantly contributed to improving the clarity and quality of our work. We trust that these revisions address your concerns and enhance the overall presentation of our research.

Thank you for your attention and valuable guidance.

Sincerely,

Takashi Honda, MD, PhD

Department of Gastroenterology and Hepatology, Nagoya University Graduate School of Medicine, Nagoya 466-8560, Japan

Tel.: +81-52-744-2169, Fax: +81-52-744-2178

E-mail: honda@med.nagoya-u.ac.jp